# Targeting of Glucose Transport and the NAD Pathway in Neuroendocrine Tumor (NET) Cells Reveals New Treatment Options

**DOI:** 10.3390/cancers15051415

**Published:** 2023-02-23

**Authors:** Jochen Winter, Rudolf Kunze, Nadine Veit, Stefan Kuerpig, Michael Meisenheimer, Dominik Kraus, Alexander Glassmann, Rainer Probstmeier

**Affiliations:** 1Oral Cell Biology Group, Department of Periodontology, Operative and Preventive Dentistry, University Hospital, Medical Faculty, University of Bonn, Welschnonnenstr. 17, 53111 Bonn, Germany; 2Neuro- and Tumor Cell Biology Group, Department of Nuclear Medicine, University Hospital, Medical Faculty, University of Bonn, Venusberg-Campus 1, 53127 Bonn, Germany; 3Department of Nuclear Medicine, University Hospital, Medical Faculty, University of Bonn, Venusberg-Campus 1, 53127 Bonn, Germany; 4Department of Prosthodontics, Preclinical Education, and Material Sciences, University Hospital, Medical Faculty, University of Bonn, Welschnonnenstr. 17, 53111 Bonn, Germany; 5Department of Immunology and Cell Biology, University of Applied Science Bonn-Rhein-Sieg, Campus Rheinbach, von-Liebig-Str. 20, 53359 Rheinbach, Germany

**Keywords:** fasentin, glucose uptake inhibitor, GMX1778, neuroendocrine, STF-31, WZB117

## Abstract

**Simple Summary:**

Compounds interfering with glucose uptake and NAD metabolism are potential candidates for cancer therapy. Tumor cells are sensitive to such compounds in terms of glucose uptake as well as cell proliferation and survival. We have carried out these types of studies to neuroendocrine tumor (NET) cells, i.e., pancreatic Bon-1 and QPG-1 NET cell lines and GLC-2 and GLC-36 small cell lung cancer (SCLC) cell lines. Cells treated with substances that provoke a chemical inhibition of facilitative glucose transporters (GLUT) and nicotinamide phosphoribosyltransferase (NAMPT), a key enzyme in NAD metabolism, lead to decreased cell proliferation and increased cell death.

**Abstract:**

(1) Background: the potency of drugs that interfere with glucose metabolism, i.e., glucose transporters (GLUT) and nicotinamide phosphoribosyltransferase (NAMPT) was analyzed in neuroendocrine tumor (NET, BON-1, and QPG-1 cells) and small cell lung cancer (SCLC, GLC-2, and GLC-36 cells) tumor cell lines. (2) Methods: the proliferation and survival rate of tumor cells was significantly affected by the GLUT-inhibitors fasentin and WZB1127, as well as by the NAMPT inhibitors GMX1778 and STF-31. (3) Results: none of the NET cell lines that were treated with NAMPT inhibitors could be rescued with nicotinic acid (usage of the Preiss–Handler salvage pathway), although NAPRT expression could be detected in two NET cell lines. We finally analyzed the specificity of GMX1778 and STF-31 in NET cells in glucose uptake experiments. As previously shown for STF-31 in a panel NET-excluding tumor cell lines, both drugs specifically inhibited glucose uptake at higher (50 μM), but not at lower (5 μM) concentrations. (4) Conclusions: our data suggest that GLUT and especially NAMPT inhibitors are potential candidates for the treatment of NET tumors.

## 1. Introduction

Neuroendocrine tumors (NETs) derive from neuroendocrine cells that in turn are descendants of the embryonic neural crest or the endodermal layer [1,2]. Inter alia, LeDouarin and coworkers have established in their classical experiments based on the replacement of embryonic chick neural crest by embryonic quail neural crest that ganglion cells of the submucous and enteric plexus of the gastrointestinal tract as well as paraganglia and thyroid C cells are of neural crest origin, whereas cells of the diffuse gastrointestinal endocrine system are most likely derived from local endoderm [1,3]. Although NETs share some common features such as the expression of certain neuroendocrine markers, they exhibit a considerable heterogeneity with respect to the site of origin, functional status, and grade of aggressiveness [4]. In a WHO reclassification from 2010, NETs have been classified according to histopathological features and proliferation characteristics, i.e., into well differentiating neuroendocrine tumors (WNETs) and poorly differentiating neuroendocrine carcinomas (PNECs), whereby the latter ones were further subdivided into small and large cell histological subgroups [5]. Additional classification systems, for example the staging of gastroenteropancreatic NETs according to their proliferation pattern have been introduced [5]. NETs have also been classified according to their site of origin and functional state [5]. Presently, an overall accepted system of nomenclature, grading, and staging of NETs is still missing, partially due to accumulating evidence that WNETs and PNECs represent distinct families of neoplasms [4]. Moreover, assuming that poorly differentiated histology and high tumor grade are equivalent has been questioned in several studies [6].

Dependent on the tumor type and stage different treatment strategies are available for NETs, comprising surgery, chemotherapy (as the application of the multi-tyrosine kinase inhibitor sunitinib and the mTOR inhibitor everolimus in metastatic pancreatic NETs) or peptide receptor radionucleotide therapy (PRRT) (as the usage of radiolabeled somatostatin analogs in well differentiated NETs) [7]. Recently, a benefit of somatostatin analog PRRT in small cell lung carcinoma (SCLC) therapy has also been reported [8].

To our knowledge, the impact of drugs interfering with glucose uptake and NAD metabolism has not been extensively studied in NETs except a study performed by Elf and colleagues (2017). These authors demonstrated that NAMPT inhibitor GMX1778 enhances the efficacy of ^177^Lu-DOTATATE treatment in nude mice that had been xenografted with the human neuroendocrine cell line GOT1 [9]. A main reason why glucose uptake inhibitors were out of scope so far is probably due to the fact that in slow-dividing non-aggressive NETs in general no enhanced glucose uptake takes place [10]. However, a subset of aggressive NETs is, inter alia, characterized by an increased glucose uptake. When ^18^F-FDG uptake was studied in patients with early stage pulmonary NETs, high ^18^F-FDG uptake correlated with high-grade malignancies and GLUT-1 expression as well as a poor outcome after curative surgery [11]. Additionally, a previous report focusing on small cell lung cancer (SCLC) patients demonstrated that increased ^18^F-FDG uptake as measured by positron emission tomography (PET) is associated with poor prognosis in limited and extensive stage disease [12].

It has been suggested that a combination of somatostatin-receptor (SSTR)-specific [13,14] and ^18^F-FDG [15] PET/computer tomography (CT) imaging may lead to better risk estimation of NET patients thus allowing the generation of alternative treatment options such as a combination of PRRT and radio-sensitizing (chemo-) therapies [16]. Compounds interfering with glucose uptake and NAD metabolism are likely candidates for such combinatorial treatments. We have recently demonstrated [17] that tumor cells representing major human tumor entities are sensitive to such compounds in terms of glucose uptake as well as cell proliferation and survival. Here we have extended these types of studies to NET cells, i.e., pancreatic Bon-1 and QPG-1 NET cell lines [18] and GLC-2 and GLC-36 SCLC cell lines [19]. We show that cells treated with metabolism interfering drugs that provoke a chemical inhibition of facilitative glucose transporters (GLUT) and nicotinamide phosphoribosyltransferase (NAMPT) lead to decreased cell proliferation and increased cell death.

## 2. Materials and Methods

### 2.1. Materials

Fasentin, WZB117, and STF-31 were from Merck Millipore. GMX1778 was obtained from Selleck Chemicals and nicotinic acid (NA) from Sigma. [^18^F]-fluoro-deoxyglucose ([^18^F]-FDG; in 0.9% sodium chloride) was kindly supplied by Advanced Accelerator Applications (Bonn, Germany).

### 2.2. Cell Lines and Culture Conditions

Human SCLC cell lines GLC-2 and GLC-36 [19], pancreatic NET (pNET) cell lines BON-1 and QPG-1 [20], glioblastoma cell line A172 and colon carcinoma cell line HT-29 [17] were maintained in DMEM, 10% fetal calf serum (FCS). Under experimental conditions, serum concentration was reduced to 1% or 0% (glucose uptake experiments only).

### 2.3. Total RNA Isolation, cDNA Synthesis, and Quantitative RT-PCR Analysis

Synthesis of cDNA was carried out after total RNA isolation as described previously [17]. Levels of expression of all genes of interest were examined by real-time PCR using the CFX Connect™ Real-Time PCR System (Bio-Rad Laboratories, Munich, Germany), SYBR^®^ Green (Bio-Rad Laboratories), with the use of specific primers (Table 1). All primers were verified by computer analysis for specification (BLAST) and chemically synthesized of high quality (Metabion, Martinsried, Germany). Primer sequences, annealing temperatures and efficiencies are designated in Table 1.

Quantitative PCR was carried out as recently described [21]. The ∆∆Ct-method described by Pfaffl [22] was used for relative differential gene expression analysis with β-actin and glyceraldehydephosphate-dehydrogenase (GAPDH) as house-keeping genes, and standardized to their mean values. Real-time PCR was carried out according to “The MIQE Guidelines: Minimum Information for Publication of Quantitative Real-Time PCR Experiments” [23]. PCR products were electrophoretically separated on 1% agarose gels and visualized with ethidium bromide, isolated, and DNA-sequenced for verification.

### 2.4. Cell Proliferation and Viability Assays

Cell viability and proliferation was determined using crystal violet and LDH assays: (1) 1000 to 5000 cells were seeded per well in 96-well plates and cultured overnight in normal culture medium. Medium was then changed to DMEM containing 1% FCS for up to four days in the absence or presence of experimental compounds in the crystal violet assay. Cells were then fixed in 5% formaldehyde in phosphate-buffered saline (PBS) for 15 min and stained for 1 h with 0.05% crystal violet in Aqua dest., subsequently washed twice with Aqua dest., and air-dried. 150 µL of methanol were added per well and the optical density at 540 nm was measured. (2) The LDH assay was carried out as described previously with the “LDH Cytotoxicity Assay Kit” from Roche (Basel, Switzerland) [19]. No drug-induced cell toxicity was observed at the concentrations used in our study.

### 2.5. Glucose Uptake Assay

For glucose uptake assays 1.5 × 10^5^ cells per well of 24 well plates were incubated overnight in complete culture medium. Experimental compounds were preapplied in a final volume of 400 µL for 30 min in glucose-free medium. Subsequently, 370 kBq, corresponding to 10 µCi (equal to 150 to 300 pmol) of [^18^F]-FDG (specific activity of 2.5 to 5.0 GBq/µmol) were added in 100 µL per well for 30 min. Afterwards, radioactive FDG containing medium was discarded and cells were rinsed twice with PBS and subsequently solubilized in PBS containing 2% Triton X-100. Glucose uptake was detected by measuring radioactivity using a gamma-counter. Values were matched according to the decay of fluorine-18.

### 2.6. Statistics

For statistical data analysis, the one-way-ANOVA and the posthoc Tukey’s multiple comparison test were used with GraphPad Prism 6 Software (San Diego, CA, USA). *p*-values < 0.05 were considered to be statistically significant.

## 3. Results

### 3.1. Proliferation and Survival Capacity of NET Cells Is Reduced in the Presence of GLUT or NAMPT Inhibitors in a Concentration Dependent Manner

NET cell lines Bon-1, QPG-1 (pNET cell lines), or GLC-2 and GLC-36 (SCLC cell lines) were cultivated for four days in the presence of increasing concentrations of fasentin, GMX-1778, STF-31, or WZB to analyze cell proliferation and survival. In all cell lines, a strong impact of fasentin (Figure 1A) became obvious only at the highest concentrations of 50.0 and 100.0 µM. At intermediate concentrations, GLC-2 cells were partially affected. The impact of WZB–117 was comparable to the one of fasentin (Figure 1D). STF-31 treatment led to a strong impact in GLC-2 and QPG-1 cells, even at concentrations as low as 0.1 µM (Figure 1C). GLC-36 cells were partially resistant to STF-31 treatment already at intermediate concentrations, whereas high resistance was observed in Bon-1 cells. Comparable data were obtained for GMX-1778-treated NET cells (Figure 1B).

In this context, we analyzed the expression level of NAMPT and NAPRT in NET cells (Figure 2). Whereas NAMPT expression was detectable in all NET cells (Figure 2A), NAPRT expression was suppressed or almost absent in two (GLC-2 and QPG-1) out of the three NET cell lines tested (Figure 2B). Unfortunately, GLC-36 cells could not be included in this type of analysis, as, by unknown reasons, no measurable or replicable values could be obtained. In control experiments, the expression level of NAMPT and NAPRT was investigated in A172, MG-63, HeLa, and HT-29 cells. As previously shown (17) NAMPT was expressed in all four cell lines, whereas NAPRT was expressed in HeLa and HT-29 cells, but almost completely diminished in A172 and MG-63 cells (Figure 2).

We next tried to rescue the effect of NAMPT inhibition via the addition of nicotinic acid. Supplementation with NA allows NAD+ generation only when the Preiss–Handler salvage pathway is intact (for further details, see Kraus et al., 2018 and references therein) and, in particular, needs the expression of functional NAPRT [24]. In all four NET cell lines tested, the inhibitory effect of the NAMPT inhibitors could not be rescued by nicotinic acid (Figure 3). These data hint at the absence of functional NAPRT or the presence of nonfunctional NAPRT, i.e., the presence of an inactive Preiss–Handler salvage pathway that does not allow NAD+ generation [24]. In accordance with previous results [17], the presence of nicotinic acid rescues HT-29 but not A172 cells. As the amount of relative drug-dependent inhibition varied in the different experiments to a considerable amount (see SD values in Figure 3A) we also present the summarized absolute percentages of the rescue effect for each experiment (GMX- or STF- versus GMX+NA or STF+NA treated-cells) in Figure 3B. Especially the second way of representing our data documents the absence of a nicotinic acid-mediated rescue effect in NET cells.

### 3.2. Glucose Uptake Is Inhibited in NET Cells in a Concentration- and Drug-Dependent Manner

We have previously shown that the GLUT-1/NAMPT inhibitors used in the present study interfere with Glucose-uptake in a drug- and cell-line-specific manner [17]. To expand these type of studies, NET cells were first treated with [^18^F]-FDG in the absence of inhibitors to analyze glucose uptake efficiency. After an incubation time of 30 min with [^18^F]-FDG in serum- and glucose-free medium, the uptake rate varied by a maximum of about factor two between the cell lines, but depicted no statistical difference (not shown). These kinds of variations are also not reflected by differences in the expression level of GLUT1 transcripts as revealed by real time RT-PCR analysis (not shown).

We next examined if GLUT or NAMPT inhibitors interfered with the cellular uptake of glucose (Figure 4).

Fasentin (100 µM) inhibited glucose uptake in the range of 30–40%, with the exception of GLC-2 cells where an inhibition rate around 70% was observed. Whereas 50 µM GMX1778 led to a pronounced inhibition of glucose uptake in the range of 90%, no inhibition took place at a concentration of 5 µM as already shown for other types of tumor cells [17]. In all cell lines 50 µM STF-31 inhibited glucose uptake significantly in the range of 25 up to around 50%. A strong effect was provoked by WZB-117 (50 µM), which reduced the glucose uptake rate with a minimum of 80%. 

## 4. Discussion

Tumor cells possess particular metabolic properties adapted to cancer-specific features, especially regarding replicative immortality [25]. Dysregulation of proliferation causes the need to provide tumor cells with enough anabolic (nucleotides, amino acids, etc.) and also usable energy (ATP) metabolites to execute cell division. In cancer, ATP is mainly produced via the Warburg effect, leading to only two molecules of ATP per molecule glucose, which is only 6% compared to full glucose metabolization to CO_2_ via the tricarboxylic acid cycle [26]. Hence, tumor cells show a high requirement for glucose making GLUTs a potential tool for targeting anticancer therapy. Another key molecule which plays an extraordinary role in cell metabolism is NAD serving as redox partner as well as a metabolic substrate in various cellular processes, e.g., proliferation or DNA damage repair. NAMPT is the rate-limiting enzyme in the NAD biosynthetic pathway, showing an overexpression in many cancer cells associated with poor prognosis. Resistance to medication treatment of cancer cells is affected by NAMPT inhibitors in such way that therapeutic decline is prolonged [27,28]. This effect might be caused by inhibiting drug metabolizing NADPH consuming phase I enzymes, such as Flavin-dependent and/or cytochrome P450 monooxygenases [17,29,30]. Besides its role as redox partners in many metabolic reactions, NAD is also used as a substrate—e.g., by ADP-ribosyl transferases and ADP-ribosyl polymerases, among others—leading to its degradation, and thus causing a lack of NAD if it is not constantly biosynthetically regenerated. Hence, NAMPT as the most important enzyme in NAD biosynthesis has also become into focus as promising target for specific cancer treatment. Inhibitors for GLUT and NAMPT have been examined as anti-tumor agents for some time already [31]. Recently, we have published that the GLUT inhibitor STF-31 shows a dual mode of action: beside its effect on glucose uptake STF-31 is also able to affect the NAD pathway as NAMPT inhibitor, although particularly at even lower concentrations [17]. Moreover, Wang and coworkers [32] have previously reported that STF-31inhibits glucose uptake after a one-hour treatment in murine microglial cells. Unfortunately, detailed molecular docking studies on the interaction of STF-31 with Glut-1 or NAMPT are missing [33]. Therefore, at present, it has to be assumed for STF-31 that its favored mode of action depends on the gene expression level of GLUT1 and NAMPT in tumor cells [17]. The first step in glucose metabolism after its entry into the cell is an ATP-dependent phosphorylation catalyzed by glucokinase [34]. The ATP level is indirectly dependent on NAD since this cofactor is required as redox partner in the lactate dehydrogenase reaction driving the anaerobic glycolytic pathway (Warburg effect) which is predominant ATP generating process in tumor cells [26]. Furthermore, a reduced cellular level of NAD can impact the enzyme NAD kinase which produces NADP, an important player in the pentose phosphate pathway. Decreased NADP levels lead to reduced biosynthesis of nucleotides [35]. Hence, glucose and NAD metabolisms are connected to each other.

The GLUT inhibitors fasentin and WZB117 showed similar effects on neuroendocrine and small cell lung cell tumor cells: both substances negatively affected the cell numbers only at higher concentrations (50/100 µM). This observation is in good agreement with results of our group on cell lines of different tumor entities. Although the cellular response varied between cell lines after treatment with those GLUT inhibitors, a dramatic effect was shown only at higher concentrations (50/100 µM) [17]. Remarkably, GLC-36 is more susceptible to WZB117 with already 5 µM. This observation is exclusive for this cell line. Such high susceptibility against WZB117 could not be shown for the other cells in this nor in our recent study [17]. Treatment with the inhibitors STF-31 is most effective at higher concentrations (50/100 µM) which is in good agreement with results on other cell lines [17]. Notably, Bon-1 has shown no significant negative impact by STF-31 stimulation. We suppose this to be due to an active detoxification machinery [29,30]. GMX-1778 is highly effective in reducing the cell numbers of GLC-2, GLC-36, and QPG-1 at even low concentrations from up to 0.1 µM but has no significant impact on Bon-1 not even at highest concentrations. The latter observation is remarkable since no cell line examined in this or our recent study [17] was resistant to GM1778 at 100 µM. A possible explanation might be a highly efficient drug metabolizing system [29,30]. Another possible reason for the resistance of Bon-1 to NAMPT inhibitors can be the fact that the expression level of NAPRT in Bon-1 cells is much higher compared to all other investigated (Figure 2). Thus, the main biosynthetic pathway producing NAD might be carried out through NAPRT, the Preiss–Handler pathway, and thereby circumvent the NAMPT pathway [36]. This explanation is supported by the results regarding rescue effects by additional treatment with NAMPT inhibitors and the precursor nicotinic acid (Figure 3). Two cell lines, namely Bon-1 and HT-29, show merely rescue effects with nicotinic acid. Additionally, both cell lines exhibit a relatively high expression level of NAPRT.

Cellular glucose uptake is reduced after treatment of Bon-1, GLC-2, GLC-36, and QPG-1 by all four substances. The inhibition profile between each cell line does not differ significantly. The results are comparable with data from our recent study [17]. Stimulation with a high concentration (50 µM) of the NAMPT inhibitor GMX1778 shows also a strong decreased glucose uptake. Glucose uptake is a primary active transport [37]. Thus, the first step after entering the cell is the phosphorylation of glucose by glucokinase to keep the glucose, as glucose-6-phosphate, within the cell [32]. This process requires ATP as cosubstrate. As mentioned above, a sufficient level of ATP is dependent on NAD. Thus, a reduced amount of NAD leads to a diminished ATP level which consequentially causes a decrease in intracellular glucose concentration.

## 5. Conclusions

To summarize, GLUT and NAMPT inhibitors negatively affect cell numbers of NET cells. Resistance to NAMPT inhibitors depends on the expression level of NAPRT and thus on the predominant NAD producing Preiss–Handler pathway. Finally, our data suggest that GLUT and especially NAMPT inhibitors are potential candidates for the treatment of NET tumors.

## Figures and Tables

**Figure 1 cancers-15-01415-f001:**
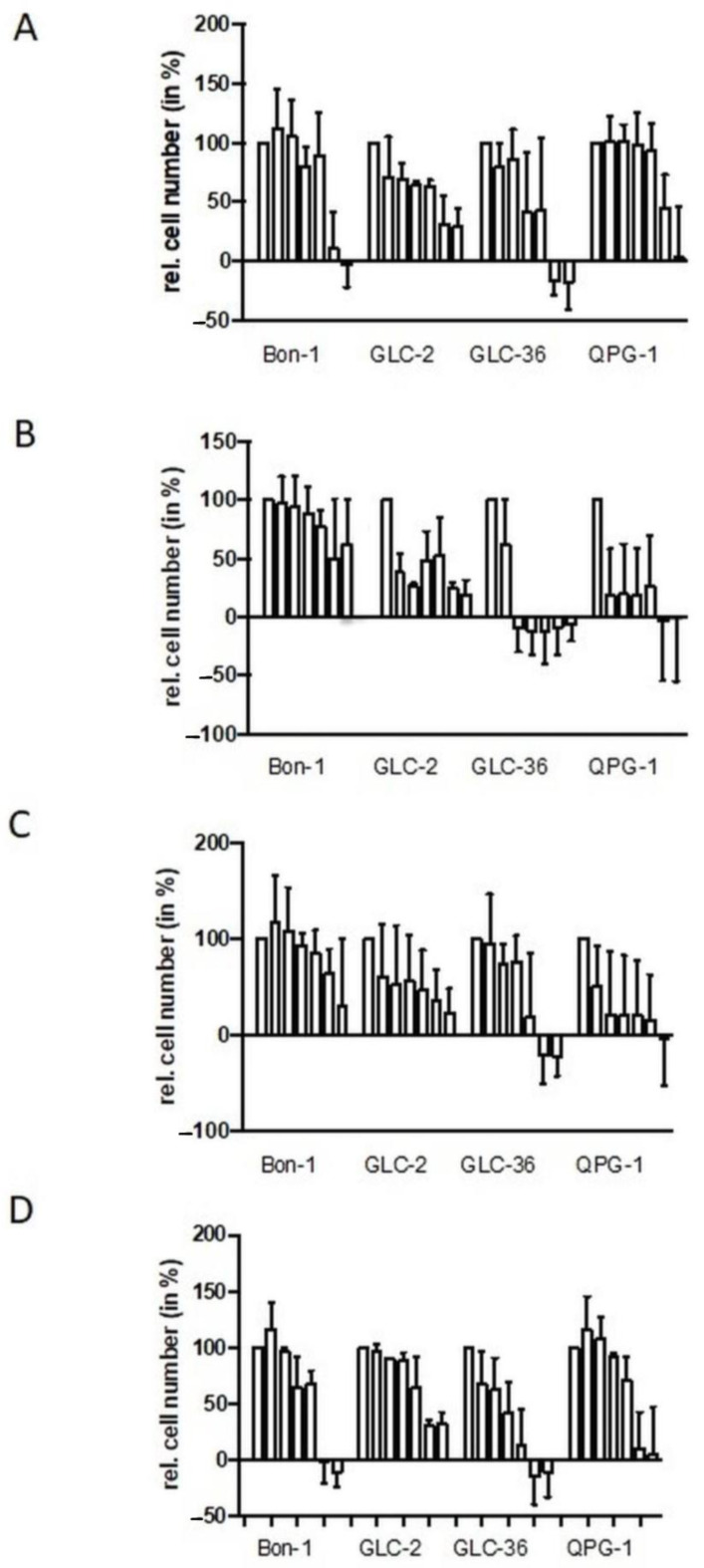
Perturbation of proliferation by GLUT and NAMPT inhibitors in NET cells. Human NET cells Bon-1, GLC-2, GLC-36, and QPG-1 were cultivated in 96-well plates for four days in the presence of 0.0, 0.1, 1.0, 5.0, 10.0, 50.0, and 100.0 µM fasentin (**A**), GMX-1778 (**B**), STF-31 (**C**), or WZB117 (**D**). Cell number was determined by crystal violet assay. Data are means + SEM from two independent experiments and relative values are given in respect to the one of untreated cells (first bar in each group of seven which was set as 100%). Negative values represent cell numbers lower than the number of seeded cells and, thus, clearly indicate cell death.

**Figure 2 cancers-15-01415-f002:**
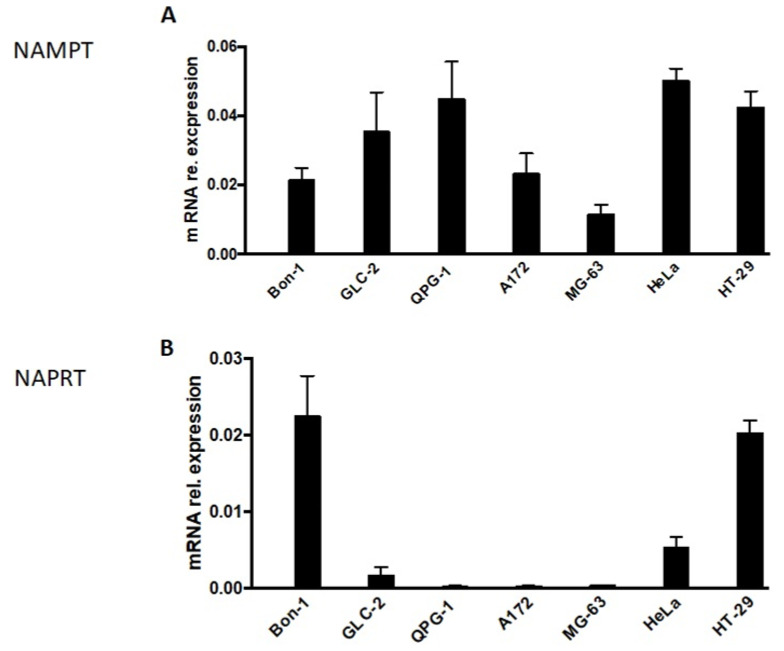
NAMPT and NAPRT expression in NET cells. NAMPT (**A**) and NAPRT (**B**)-specific mRNA expression in NET cells as determined by qPCR. Data are normalized to mRNA expression of internal standard genes and given as means + SEM from two independent experiments. In comparison the expression levels in A172, MG-63, HeLa, and HT29 cells are shown. These data were adopted from [17].

**Figure 3 cancers-15-01415-f003:**
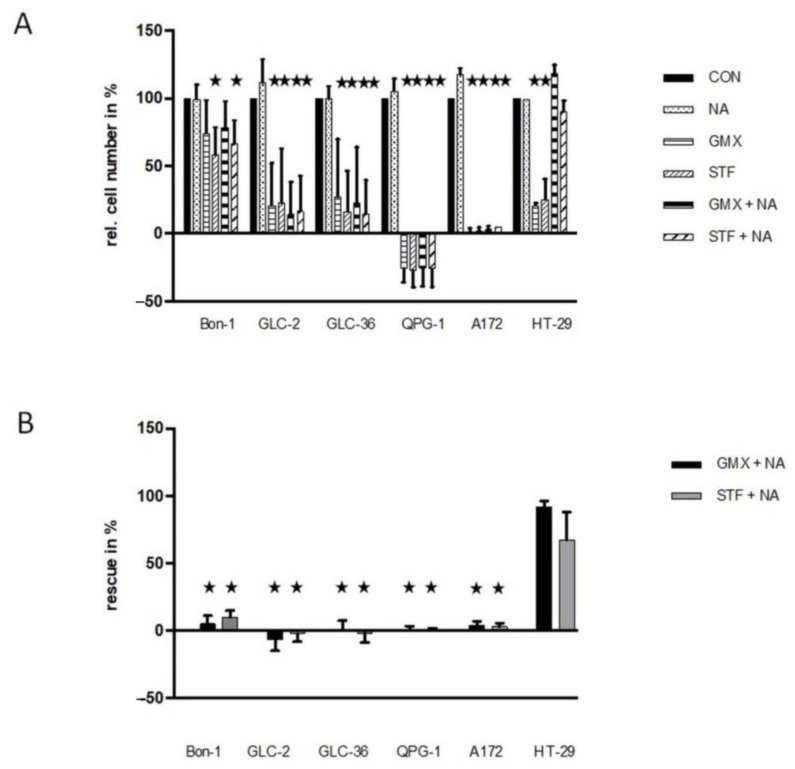
Rescue effects in NAMPT- or GLUT-inhibitor-treated NET cells. (**A**) Tumor cells were cultivated in 96-well plates for 4 days in the presence or absence of 10 µM STF-31 or 5 µM GMX-1778 in medium that was or was not supplemented with 10 µM NA. For further details, see the legend of Figure 1. In subfigure (**B**), the relative rescue efficiency is shown in %. Optimal non-toxic concentrations as revealed in Figure 1. Asterisks (★) show statistically significant differences (*p* < 0.05) compared to untreated controls.

**Figure 4 cancers-15-01415-f004:**
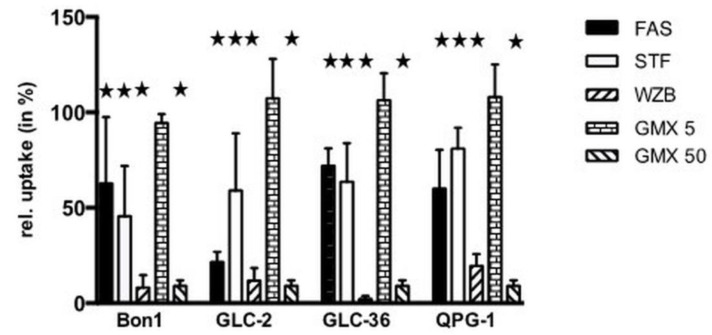
Inhibition of glucose uptake in NET cells treated with NAMPT- and Glut-inhibitors. Tumor cells were incubated for 30 min (a) either in the absence or presence of 100 µM fasentin (FAS), 50 µM STF-31 (STF), 50 µM WZB117 (WZB), 5 or 50 µM GMX-1778 (GMX). The uptake of [^18^F]-FDG in the absence of inhibitors was set at 100% for each cell line. Corresponding values of cell treatment in the presence of compounds are shown. Asterisks (★) indicate statistically significant differences (*p* < 0.05) to untreated controls.

**Table 1 cancers-15-01415-t001:** Genes, primer sequences, corresponding efficiencies, annealing temperatures, and PCR product lengths.

Gene	Primer Sequences (Sense/Antisense)	Efficiency	Annealing Temperature (°C)	Product Length (bp)
β-actin	5′-CATGGATGATGATATCGCCGCG-3′ 5′-ACATGATCTGGGTCATCTTCTCG-3′	1.84	69	371
*GAPDH*	5′-TGGTATCGTGGAAGGACTCA-3′ 5′-CCAGTAGAGGCAGGGATGAT-3′	1.93	67	132
*NAMPT*	5′-CTTCTGGTAACTTAGATGGTCTGGAA-3′ 5′-TGCTCCTATGCCAGCAGTCTCTT-3′	1.91	66	89
*NAPRT*	5′-CAGGTGGAGCCACTACTGC-3′ 5′-CGTGTTGTTTCCAGTCAGCC-3′	2.06	69	245

## Data Availability

Supporting data and materials are available on request to the corresponding author.

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
