# Peer review of "Targeting of Glucose Transport and the NAD Pathway in Neuroendocrine Tumor (NET) Cells Reveals New Treatment Options"

_cancers, 2023, doi:10.3390/cancers15051415_

Round 1
Reviewer 1 Report
Winter et al have worked on employing inhibitors of glucose uptake and NAD metabolism for NET therapy. The authors explore highly aggressive NET samples as they have high glucose uptake and therefore would be more susceptible to glucose uptake inhibitors. Their work shows that inhibitors targeting GLUT and NAMPT affects cell proliferation and cell death. While this manuscript seems to have gone over mostly preliminary experiments it would be nice to see this work develop towards clinically relevant NET therapy systems.
There are a few opportunities to improve the impact as well as the clarity of the manuscript-
Page 6 Line 174 - Why did the authors not use Western Blots in addition to qPCR to visualize the protein expression of GLUT1, NAMPT, NAPRT in different cell lines? Is there a reason the authors did not look into GLUT1 expression? It would be helpful to see this data in supplementary data.
Page 7 Line 198-204 – Please explain this with a little more details as the message in not very clear
Figure 3 – I am a little confused when the author refers to legend of figure 1 as there each bar represented a different concentration of the inhibitor. Vs here in Figure 3 each bar represents a different set of conditions. Please clarify this. Also please share the concentrations of each condition/inhibitor in the legend.
Please proof read the manuscript to correct typos in the manuscript.
General comments to increase the impact of the manuscript-
Including any in vitro inhibitory assays with the inhibitors would be helpful as positive/negative controls.
In vivo animal studies were to show tumor shrinkage would be significantly increase the impact of the manuscript.
The authors should add the results of cytotoxicity studies to the manuscript or supplemental data to help the readers.
Instead of only using cell lines available with innate expression of GLUT1, NAMPT, and NARPT, have the authors thought of synthesizing cell lines that over express these proteins to show that by modulating the protein expression they could vary the resistance/susceptibility towards the inhibitors?
The authors hint toward using glucose uptake and NAD metabolism inhibitors as good candidates for combinatorial treatment. Showing data for the same would be helpful in increasing the impact of the manuscript.
Author Response
Comments to reviewer
Dear reviewer,
we are very grateful for all your comments and suggestions which we really appreciate. They were very helpful and have definitely improved our manuscript.
Comments and Suggestions for Authors
Winter et al have worked on employing inhibitors of glucose uptake and NAD metabolism for NET therapy. The authors explore highly aggressive NET samples as they have high glucose uptake and therefore would be more susceptible to glucose uptake inhibitors. Their work shows that inhibitors targeting GLUT and NAMPT affects cell proliferation and cell death. While this manuscript seems to have gone over mostly preliminary experiments it would be nice to see this work develop towards clinically relevant NET therapy systems.
There are a few opportunities to improve the impact as well as the clarity of the manuscript-
Page 6 Line 174 - Why did the authors not use Western Blots in addition to qPCR to visualize the protein expression of GLUT1, NAMPT, NAPRT in different cell lines? Is there a reason the authors did not look into GLUT1 expression? It would be helpful to see this data in supplementary data.
Our comments: We agree with the reviewer that protein expression should have been analyzed. Unfortunately, we could not find any reliable antibodies in respect to sensitivity and specificity that could be used.
Page 7 Line 198-204 – Please explain this with a little more details as the message in not very clear
Our comments: We have replaced now „As the amount of relative drug inhibition……were summarized (Fig. 3B)“ by „As the amount of relative drug-dependent inhibition varied in the different experiments to a considerable amount (see SD values In Fig. 3A) we also present the summarized absolute percentages of the rescue effect for each experiment (GMX- or STF- versus GMX+NA or STF+NA treated-cells) in Fig. 3B.”
Figure 3 – I am a little confused when the author refers to legend of figure 1 as there each bar represented a different concentration of the inhibitor. Vs here in Figure 3 each bar represents a different set of conditions. Please clarify this. Also please share the concentrations of each condition/inhibitor in the legend.
Our comments: Optimal non-toxic concentrations as revealed in Fig. 1. We have clarified by modification of legend of Figure 3.
Please proof read the manuscript to correct typos in the manuscript.
Our comments: thank you. We have done it.
General comments to increase the impact of the manuscript-
Including any in vitro inhibitory assays with the inhibitors would be helpful as positive/negative controls.
Our comments: We have focused in our manuscript on cell proliferation/survival and glucose uptake at non-toxic inhibitor concentration. In this context we did not plan to integrate other cellular parameters in our study.
In vivo animal studies were to show tumor shrinkage would be significantly increase the impact of the manuscript.
Our comments: The application of the drugs used in our study (and also other ones that are available in in vivo experiments seems to be critical, as they are relatively fast inactivated in the liver. So, drugs are urgently needed that are more stable in vivo.
The authors should add the results of cytotoxicity studies to the manuscript or supplemental data to help the readers.
Our comments: Data demonstrating that the drugs used are non-toxic at the desired concentrations are a prerequisite of the whole study and, thus, in our opinion, do not legitimate a separate figure. We have mentioned now our results in M&M (line 138): “No drug-induced cell toxicity was observed at the concentrations used in our study.“
Instead of only using cell lines available with innate expression of GLUT1, NAMPT, and NARPT, have the authors thought of synthesizing cell lines that over express these proteins to show that by modulating the protein expression they could vary the resistance/susceptibility towards the inhibitors?
Our comments: We have not planned to perform such types of experiments.
The authors hint toward using glucose uptake and NAD metabolism inhibitors as good candidates for combinatorial treatment. Showing data for the same would be helpful in increasing the impact of the manuscript.
Our comments: The reviewer is right that such experiments would be helpful. However, such type of experiments would be very time-consuming, i.e. at least six months, and break the limits of our study.
Reviewer 2 Report
The paper entitled "Targeting of Glucose Transport and the NAD Pathway in Neuroendocrine Tumor (NET) Cells Reveals New Treatment Options" showed the combination effect of GLUT-inhibitors fasentin and WZB1127, as well as by the NAMPT 32 inhibitors GMX1778 and STF-31.
the paper is good but there is some concerns.
-the lack of toxicity study of this combination on normal neuroendocrine cell lines as well as small cell lung
-the compounds tested are correlated to different chemical structure which make the mode of interaction between them to receptors not clear
so I suggest running Molecular docking study to show the interaction as well as molecular dynamics
Author Response
Comments to reviewer
Dear reviewer,
we are very grateful for all your comments and suggestions which we really appreciate. They were very helpful and have definitely improved our manuscript.
Comments and Suggestions for Authors
The paper entitled "Targeting of Glucose Transport and the NAD Pathway in Neuroendocrine Tumor (NET) Cells Reveals New Treatment Options" showed the combination effect of GLUT-inhibitors fasentin and WZB1127, as well as by the NAMPT 32 inhibitors GMX1778 and STF-31.
the paper is good but there is some concerns.
-the lack of toxicity study of this combination on normal neuroendocrine cell lines as well as small cell lung
Our comments: To our knowledge no normal human neuroendocrine cell line is available. There is also no protocol available that allows the isolation of primary human lung-derived neuroendocrine cells (beside of ethical reasons).
-the compounds tested are correlated to different chemical structure which make the mode of interaction between them to receptors not clear
Our comments: The reviewer is right, that detailed studies on drug-receptor interactions are missing on the molecular level. Thus, we are confronted with the analysis of the effects on the cellular level. Such a strategy seems to be quite “normal” for most drugs introduced and used in biomedical science and molecular docking studies are out of the scope of our study, which focuses on drug-induced effects on the cellular level.
Round 2
Reviewer 1 Report
The response by the authors to the comments is satisfactory.
Author Response
Dear reviewer,
thank you very much for your help.
Reviewer 2 Report
Many thanks for your reply
but still my comment that the wok still need more clarification concerning the drug receptor interaction even by in situ studies
Author Response
Dear reviewer,
we are very grateful for all your comments and suggestions which we really appreciate. They were very helpful and have definitely improved our manuscript.
Comments and Suggestions for Authors
Many thanks for your reply
but still my comment that the wok still need more clarification concerning the drug receptor interaction even by in situ studies.
Our comments: we have added the following text in the discussion section (lines 261- 269): Recently, we have published that the GLUT inhibitor STF-31 shows a dual mode of action: beside its effect on glucose uptake STF-31 is also able to affect the NAD pathway as NAMPT inhibitor, although particularly at even lower concentrations [17]. Moreover, Wang and coworkers (2019) have previously reported that STF-31 inhibits glucose uptake after a one hour treatment in murine microglial cells. Unfortunately, detailed molecular docking studies on the interaction of STF-31 with Glut-1 or NAMPT are missing (see also Adams et al., 2014). Therefore, at present, it has to be assumed for STF-31 that its favoured mode of action depends on the gene expression level of GLUT1 and NAMPT in tumor cells [17].
Thank you again for your help..
Round 3
Reviewer 2 Report
thanks for your reply